# Rapid Extraction of Viral Nucleic Acids Using Rotating Blade Lysis and Magnetic Beads

**DOI:** 10.3390/diagnostics12081995

**Published:** 2022-08-17

**Authors:** Minju Bae, Junsoo Park, Hyeonah Seong, Hansol Lee, Wonsuk Choi, Jiyun Noh, Woojoo Kim, Sehyun Shin

**Affiliations:** 1School of Mechanical Engineering, Korea University, Seoul 02841, Korea; 2Department of Micro-Nano Systems, Korea University, Seoul 02841, Korea; 3Asia Pacific Influenza Institute, Korea University College of Medicine, Seoul 02841, Korea; 4Division of Infectious Diseases, Department of Internal Medicine, Korea University College of Medicine, Seoul 02841, Korea; 5Engineering Research Center for Biofluid Biopsy, Seoul 02841, Korea

**Keywords:** virus, nucleic acid, extraction, blade, lysis, magnetic beads

## Abstract

The complex and lengthy protocol of current viral nucleic acid extraction processes limits their use outside laboratory settings. Here, we describe a rapid and reliable method for extracting nucleic acids from viral samples using a rotating blade and magnetic beads. The viral membrane can be instantly lysed using a high-speed rotating blade, and nucleic acids can be immediately isolated using a silica magnetic surface. The process was completed within 60 s by this method. Routine washing and eluting processes were subsequently conducted within 5 min. The results achieved by this method were comparable to those of a commercially available method. When the blade-based lysis and magnetic bead adsorption processes were performed separately, the RNA recovery rate was very low, and the Ct value was delayed compared to simultaneous lysis and RNA adsorption. Overall, this method not only dramatically shortens the conventional extraction time but also allows for its convenient use outside the laboratory, such as at remote field sites and for point-of-care testing.

## 1. Introduction

The current COVID-19 pandemic has caused unimaginable socio-economic burdens as well as life losses and psychological damage to all countries worldwide [1]. It has revealed many limitations in advanced science and technology in terms of practical use. Precision and rapid diagnostics that help block the spread of infectious diseases are critical for infectious disease management. Thus, diagnostic tools and methods that are cost-effective, easy to use, and highly sensitive in detecting infected subjects have been developed. However, most techniques focus on target amplification and detection without considering sample preparation, which limits the overall process time and accuracy [2]. For instance, recent studies have reported plasmonic heating-based ultrafast PCR devices, which can complete 40 thermal cycles in 10 min [3,4].

However, sample preparation, including lysis, binding, washing, and elution, requires 1–3 h in a well-equipped laboratory [3]. Due to the demand for large-scale testing, commercial RNA extraction kits and platforms have encountered various technical limitations. Furthermore, the two-year, continuing COVID-19 pandemic has resulted in major shortages of supplies, including lysis buffers and plastic pipette tips [5]. This situation triggered the search for alternative and efficient platforms for sample preparation in order to manage the next pandemic. Recently, there have been a few interesting studies on rapid viral RNA extraction using simple platforms. Using a combination of proteinase K treatment and thermal shock, SARS-CoV-2 RNA has been successfully extracted and detected [6]. In addition, a commercial kit (QIAprep&amp Viral RNA UM Kit, Qiagen, Germantown, MD, USA) combined with a liquid-based sample preparation step can facilitate viral RNA extraction in only 2 min, which is directly applicable in real-time RT-PCR detection in a streamlined workflow [7]. Moreover, a nano-plasmonic lysis technique has been successfully used to lyse pathogens, followed by photonic PCR [3].

In our previous study, we reported a new, simple method for activating platelets using a rotational stirrer in a small chamber within a minute and without activating any agonists [8]. With a precise rheological design for the stirrer, we were able to generate a sufficiently high shear stress to activate platelets in a short time. We used a shear-generating stirrer to lyse cells and pathogens < 1 μm, including exosomes and viruses. In addition, conventional lysis and RNA isolation processes were performed separately, and the required time for each process was recorded. In the present study, we integrated the rotational lysis and the magnetic bead-based two-step RNA isolation into one step. Silica-based magnetic beads and a stirrer were placed in the viral sample, and rotational shearing was applied. Using the method reported herein, one can extract viral RNA within 5 min using a micro-stirrer and magnetic beads. Hence, in this study, we demonstrate an innovative, rapid, and simple method for viral RNA extraction that could potentially make significant contributions to large-scale or single-virus tests.

## 2. Materials and Methods

### 2.1. Design of Rotating Blade System

Figure 1 shows a schematic diagram of viral RNA extraction using rotating blade lysis and silica magnetic beads. As shown in Figure 1a, the high-speed rotating stirrer provided a high shear stress flow field in the microchamber, and the stirrer acted as a blade that physically cut the membranes of the virus or exosomes. After physical lysis, the internal contents were released and the magnetic beads immediately absorbed the nucleic acids while stirring at high speed. The lysis and RNA isolation processes occurred simultaneously within 1 min, as shown in Figure 1b. It is worth noting that the conventional lysis process (Figure 1c) takes 15–60 min, followed by magnetic insertion and incubation.

The rotating blade system comprised a microchamber (500 μL) and a bar stirrer made of SS400, which could be easily magnetized upon the application of a magnetic field. The typical dimensions of a rectangular bar stirrer are 2 mm in width, 1 mm in thickness, and 7 mm in length. A cross-shaped stirrer was fabricated by adding thin wings with a width of 0.5 mm. A thin shaft placed in the center of the stirrer was in the center groove of the chamber to maintain stable rotation, even at high speeds. Hundreds of these bar stirrers can be inexpensively manufactured using an etch process, with a standard deviation of <10 μm. The rotational stirrer can be simulated using a parallel rotational disc, which is a typical type of viscometry [9]. Thus, the determination of the shear rate between the bottom surface and the rotating stirrer is straightforward. The corresponding shear rate for a given radial position in the parallel discs can be derived from the equations of motion: $\dot{\gamma }$ = 2 Rω/3 h. With the geometrical dimensions (R = 5.5 mm, h = 0.5 mm) and rotational speed (ω = 419 rad/s), the calculated shear rate was 3070 s^−1^. We also previously reported a strong secondary flow in the stirring flow within a chamber [9]. With the main and secondary flows, the biological particles would be chopped by a high-speed rotating stirrer, whose function is similar to that of the blade that cuts the viral membrane.

### 2.2. Sample Collection

Clinical samples were collected from the hospital-based influenza morbidity and mortality (HIMM) surveillance system [10]. In this study, the A/California/07/2009 (H1N1)pdm09 strain and three nasopharyngeal swab samples, for which the influenza A virus was diagnosed by RAT and RT-PCR, were used. Clinical samples were collected using the BD Universal Viral Transport Collection kit comprising 3 mL of viral transport medium (VTM) and a swab. The collected VTM was dispensed into 1.5 mL tubes and stored at −80 °C until further use. The protocol was approved by the Institutional Review Boards of the Korea University Guro and Ansan Hospital (2017GR0172 and 2017AS0418). The A/California/07/2009 (H1N1)pdm 09 strain was cultured using Madin–Darby canine kidney cells.

### 2.3. Viral RNA Extraction Using Spin Column

A spin column-based viral RNA extraction was performed according to the recommended protocol. The reagent in the QIAamp Viral RNA Mini Kit (52,904; QIAamp) was used to extract the virus. A lysate buffer (AVL) containing 1 µg carrier RNA was prepared prior to the experiment. AVL was added at a ratio of 4:1 of the required sample volume into a 15 mL conical tube. Next, 200 μL of viral media was added to the tube containing the buffer. The mixture was homogenized by vortexing for 15 s and incubated at room temperature (25 °C) for 10 min. Pure ethanol (96–100%) was added to the mixture at a ratio of 4:1 of the require sample volume. The mixture, with a total volume of 1.8 mL, was transferred to a spin column in a 2 mL collection tube, centrifuged at 6000× *g* for 1 min, and then placed into a clean 2 mL collection tube. The tube containing the filtrate was discarded. During the washing step, 500 μL of wash buffer 1 was added and centrifuged at 6000× *g* for 1 min. The column was then placed into a clean 2 mL collection tube, and the tube containing the filtrate was discarded. Next, 500 μL of wash buffer 2 was added to this tube and centrifuged at full speed (20,000× *g*; 14,000 rpm) for 3 min. In the drying step, the spin column in a 2 mL collection tube was centrifuged at full speed (20,000× *g*; 14,000 rpm) for 1 min, and the spin column was transferred to a fresh 1.5 mL elution tube. Finally, in the elution step, 60 μL of the elution buffer was carefully applied to the center of the spin column and centrifuged at 6000× *g* for 1 min.

### 2.4. Analysis of Extracted Viral RNA

The purity and quantity of the extracted viral RNA were determined by measuring the ratio of absorbance at 260 and 280 nm of 1 μL of the final eluent buffer with the use of a DS-11 FX+ spectrophotometer (Denovix, Wilmington, DE, USA). M gene, a housekeeping gene of the influenza virus, was used as the internal control in the RNA amplification. In the reverse transcription step, the extracted RNA samples were prepared using a T100 Thermal Cycler (Bio-Rad, Hercules, CA, USA) with TaqMan Reverse Transcription Reagents (Invitrogen, Waltham, MA, USA) as follows: one cycle at 25 °C for 10 min, 37 °C for 30 min, and 95 °C for 5 min. Next, cDNA was synthesized using the CFX96 Touch Real-Time PCR Detection System (Bio-Rad) as follows: one cycle at 95 °C for 10 min and 95 °C for 10 s, followed by 40 cycles at 60 °C for 30 s. The primers (FluAM-7F:5′-CTTCTAACCGAGGTCGAAACGTA-3′ and FluAM-161R: 5′-GGTGACAGGATTGGTCTTGTCTTTA-3′) were synthesized by Bioneer (Daejeon, South Korea) [11]. Twenty microliters of the sample was placed in a transparent 0.1 mL 8-tube strip (Thermo Fisher Scientific, Waltham, MA, USA) and sealed with an 8-strip cap (Applied Biosystems, Waltham, MA, USA). Data were analyzed using the Bio-Rad CFX Maestro software, and the cycle threshold (Ct) was set to 500.

### 2.5. Agarose Gelelectrophoresis

Agarose powder was dissolved in TBE buffer, and the solution was heated. The heated agarose solution was then cooled to room temperature to form a transparent gel. The agarose gel was then gently transferred to an electrophoresis chamber bathed in TBE buffer, with the gel-loading wells close to the cathode side. The samples (a mixture of 10 μL PCR product and 2 μL loading buffer) were carefully loaded into the wells. The agarose gel was subjected to electrophoresis for 1 h at a voltage of 100 V. The gels were pre-stained with GreenStar Nucleic Acid Staining Solution I (Bioneer), and a band size of 154 bp was visualized under UV light using a Gel Doc System (Bio-Rad).

## 3. Results

### 3.1. Effect of Rotational Speed of Stirrer

To verify the concept proposed in this study, we first determined whether the virus membrane had been effectively lysed by the rotation of the blade and the internal RNA had been extracted. To this end, while changing the rotational speed of the blade to 0, 1000, 2000, 3000, and 4000 rpm with a fixed rotational time of 15 min, the concentration of the RNA extracted from the virus was measured and compared to that of a control adopting QIAamp (Figure 2a). As the rotational speed of the blade increased, the concentration of the extracted RNA also increased. The best results observed at 4000 rpm were nearly identical to those of the control. It is worth noting that even at 1000 rpm, 40% of the viral membrane could be lysed. The threshold cycles (Cts) were examined by varying the rpm of the rotating blade. Similar to the RNA concentration, Ct progressively decreased with the rotational speed of the blade. The best Ct (23.5) observed at 4000 rpm was nearly identical to that of the control (23.1), and the gel electrophoresis results confirmed that the target gene was extracted from the plasma and was amplified (Figure 3b).

### 3.2. Effect of Rotation Time

After confirming the optimum rotational speed of the blade, we determined the optimal rotation period for virus lysis. For this, the rotation periods were changed to 0, 1, 3, 5, and 15 min, with a fixed rotational speed of 4000 rpm, and the extracted RNA was measured by evaluating the RNA concentrations and the Ct values. The RNA concentrations and Ct values did not differ significantly among the different rotation periods (Figure 3). These results confirm that rotating blade lysis occurred instantly, within a minute and at the optimized rotational speed. Virus lysis was completed within 1 min by rotating the blades at 4000 rpm.

### 3.3. Comparison of Two Protocols

We prepared two protocols for the rotating blade lysis and RNA extraction systems. In protocol A, cell lysis using a rotating blade was immediately followed by RNA extraction with magnetic beads (Figure 4a). Conversely, protocol B involved two steps, as shown in Figure 4b. The rotational speed was 4000 rpm, and rotation time was 1 min. 

Surprisingly, the results of the two protocols significantly differed in terms of RNA recovery and Ct values. Compared with the control and protocol A, protocol B showed inferior results. These results imply that when a long viral RNA is exposed to a blade rotating at a high speed, it may be easily cut into small pieces, which are difficult to amplify. However, as shown in Figure 4a, when magnetic beads were used with a rotating blade, viral RNA lysed from the virus membrane immediately attached to the surface of the magnetic beads and was protected from the rotating blade. Thus, the inclusion of magnetic beads in the rotating blade system can minimize the loss of extracted RNA. In addition, we examined the effect of magnetic bead concentration on viral RNA extraction in Appendix A.

### 3.4. Application to Clinical Samples

All previous experiments, whose results are presented in Figure 2, Figure 3 and Figure 4, were conducted using cultured viruses. However, it is necessary to confirm the utility of the present method using clinical samples obtained from infected patients. Using three viral samples obtained from infected patients, we conducted similar experiments (Figure 5). For all three samples, the 260/280 absorbance ratios ranged between 1.6 and 2.0, which indicates that the extracted RNA was of good purity (Figure 5a) [12]. Gel electrophoresis confirmed that the amplified RNA was the target gene with a band size of 154 bp. Furthermore, the concentrations of the extracted RNA and the Ct values of the present method were compared with that of QIAamp. As depicted in Figure 5c,d, the present method showed better results for sample number D1816, whereas the QIAamp showed better results for sample number A2887. Thus, the overall extraction performance of the QIAamp and the present method can be deemed as equivalent for clinical sample applications.

## 4. Discussion

In this study, we demonstrated the feasibility of virus lysis with the use of a rotating blade in a chamber. Based on our previous reports [9], the estimated shear rate (approximately 3000 s^−1^) applied to viral particles would not disrupt them. Due to the high shear rate, the viral membrane may be elongated, and some internal nucleic acids may leak without membrane lysis. However, the present method showed an equivalent extraction performance as compared to the chemical lysis method (control); thus, there would be serious physical lysis of the viral membrane due to the high-speed rotating blades. One can recall a conventional bead-beating mechanism that can lyse biological particles. However, in our preliminary study, it showed poor performance in lysing the viral membrane (data not shown). Thus, we conclude that the high-speed rotating blade solely contributed to the virus lysis. 

The chopping and crushing mechanism rapidly lysed cells, as shown in Figure 3. However, such a mechanism comprising a high-speed rotating blade disintegrates the nucleic acids released from the viral particles. Thus, the released nucleic acids need to be immediately protected from physical lysis. Serendipitously, we found that the magnetic beads functionalized with silica could instantly bind the released viral RNA and protect it from the high-speed rotating blade. As a result, the physical lysis and isolation steps were automatically integrated, resulting in several advantages.

The main advantage of this process is a significant reduction in the time required for complete virus lysis. As shown in Figure 1, the conventional lysis process, which took 15–60 min, was reduced to less than 1 min. Moreover, the current method does not require a chemical lysis buffer, which is currently in short supply. As previously discussed, alternative methods for efficient cell lysis that do not depend on a single reagent are required. Owing to the physical lysis mechanism, there is no need to remove the chemical buffer, including targeted nucleic acids, from the solution. In the present method, the magnetic beads can be removed from the liquid in a single step, which results in convenience and ease of operation. However, this method also has certain disadvantages. The present method requires a magnetizable blade, microchamber, and stirrer for effective physical lysis. As previously mentioned, we demonstrated the feasibility of manufacturing stirrers in large quantities inexpensively with the use of etching techniques. However, a blade stirrer can be easily made from magnetizable steel sheets by using scissors. Small tubes and rotary stirrers are easily accessible in the laboratory. 

Considering the results of clinical sample A2887 presented in Figure 5, the proposed method was somewhat inferior to QIAamp. These results might be due to some characteristic impurities in the clinical samples. In fact, most clinical samples vary greatly depending on the collection method. Since large amounts of human secretions are fed directly to the VTM, clinical samples contain more impurities, which often interfere with PCR amplification [13,14]. Therefore, to ensure the utility of the proposed technology for on-site clinical diagnosis, it is advisable to conduct an additional washing step to remove the various impurities contained in the VTM.

In conclusion, we confirmed that the present method, which involves the physical lysis of viral particles and the instant absorption of nucleic acids, could innovatively resolve the sample preparation process, which remains difficult for the whole process of virus detection. The present method innovatively shortens the sample preparation time without reducing the extraction and detection performance. It is expected to be applied in basic research laboratories and point-of-care testing that require lysis and the isolation of viral RNA or DNA from clinical samples [15]. The present method can be further extended to other biological particles such as extracellular vesicles, bacteria, fungi, and other cells.

## Figures and Tables

**Figure 1 diagnostics-12-01995-f001:**
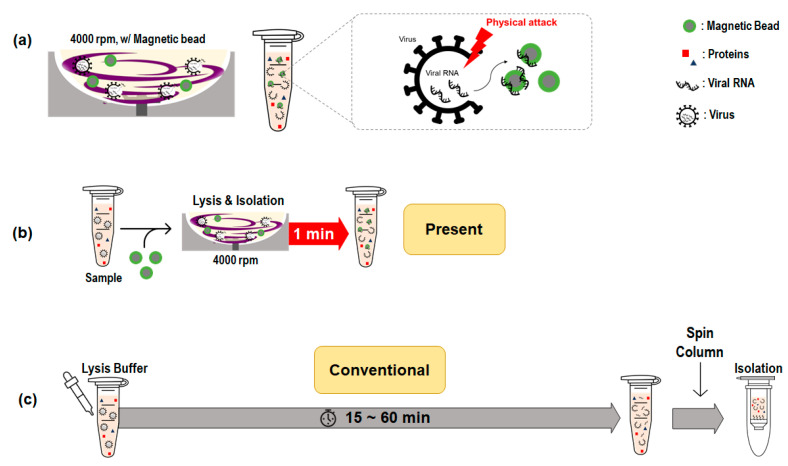
Schematic of viral RNA extraction using rotating blade lysis and silica magnetic beads. (**a**) Pathogen lysis using a high-speed rotating blade in a microwell; (**b**) Present method: Simultaneous lysis and isolation of viral RNA; (**c**) Conventional method: Stepwise lysis and spin-column extraction.

**Figure 2 diagnostics-12-01995-f002:**
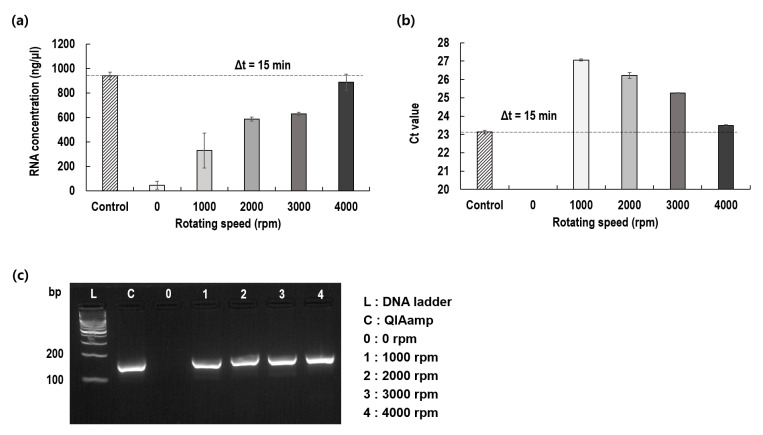
Effect of the rotational speed of the blade on viral RNA extraction as compared to control. (**a**) Extracted RNA concentrations; (**b**) Threshold cycles (Ct) of PCR for extracted RNA; (**c**) Gel electrophoresis for amplified RNA.

**Figure 3 diagnostics-12-01995-f003:**
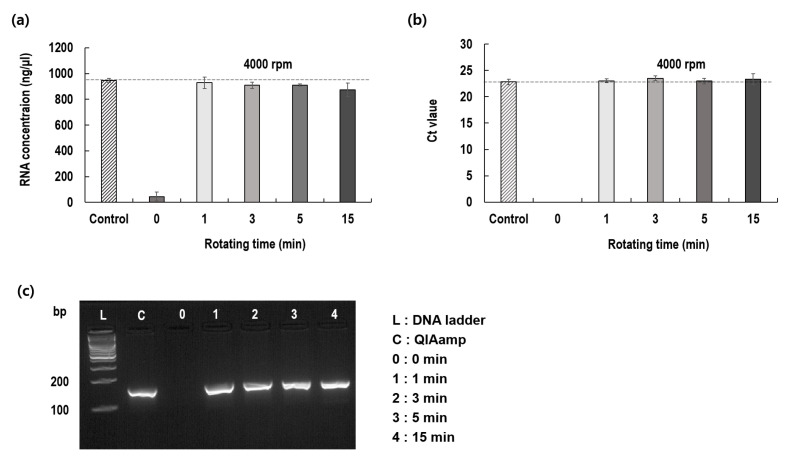
Effect of shearing time on viral RNA extraction as compared to control. (**a**) Extracted RNA concentrations; (**b**) Threshold cycles of PCR; (**c**) Gel-phoresis for amplified RNA.

**Figure 4 diagnostics-12-01995-f004:**
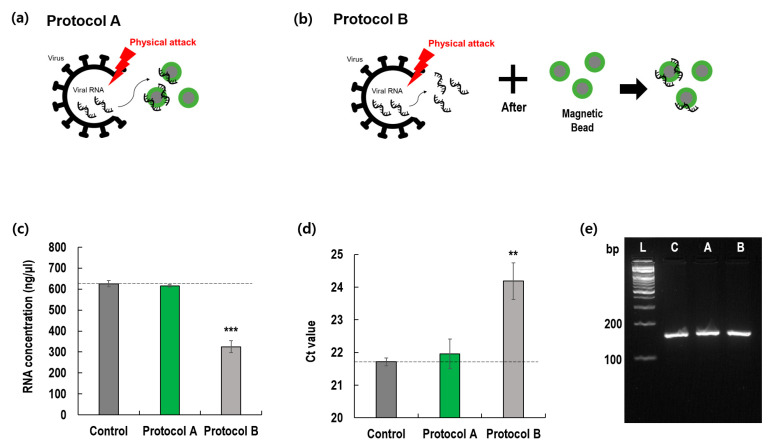
Comparison of two protocols (A and B). (**a**) Protocol A; (**b**) Protocol B; (**c**) Extracted RNA concentrations; (**d**) Threshold cycles of PCR for extracted RNA; (**e**) Gel electrophoresis of amplified RNA (** *p* < 0.005; *** *p* < 0.001) (L: DNA ladder, C: QIAamp, A: Protocol A, B: Protocol B).

**Figure 5 diagnostics-12-01995-f005:**
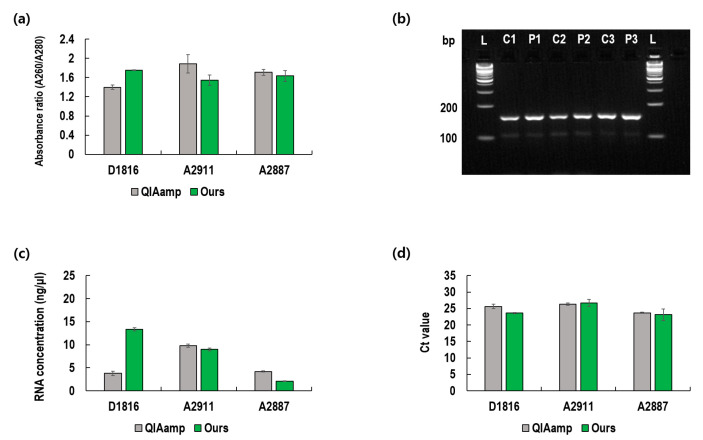
Comparison of the proposed method with the QIAamp kit using clinical samples. (**a**) Absorbance ratio as an index of purity; (**b**) Gel electrophoresis of amplified RNA (L: DNA ladder, C1: D1816 QIAamp, P1: D1816 Ours, C2: A2911 QIAamp, P2: A2911 Ours, C3: A2887 QIAamp, P3: A2887 Ours); (**c**) Extracted RNA concentrations; (**d**) Threshold cycles of PCR amplification of extracted RNA.

## Data Availability

The authors confirm that the data supporting the findings of this study are available within the article [and/or] its Appendix A.

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
