# Peer review of "Rapid Extraction of Viral Nucleic Acids Using Rotating Blade Lysis and Magnetic Beads"

_diagnostics, 2022, doi:10.3390/diagnostics12081995_

Round 1

Reviewer 1 Report

General Comments

This study ‘Rapid extraction of viral nucleic acids using rotating blade lysis and magnetic beads’ seeks to present an alternate method for RNA extraction from viral samples, as a tool for field applications. Although, an alternative approach will be a welcome tool if it is cost effective and rapid, the authors should address these concerns.

Method and Results

·        What reagent were the nasal swabs used to test the proposed system stored in?

·        A highly pure RNA sample is expected to have a 260/280 ration of 2.0, How do authors explain the seeming lack of purity in the field samples

·        How did the ratio (purity) of RNA extracted using the proposed method compare with that of the Kit?

·        It is not enough for authors to show only an agarose gel image for the field samples, it will be necessary for authors to sequence the amplified product and confirm that it is indeed influenza.

·        Do authors expect the proposed method to be applicable to other biological samples such as  blood and or arthropods?

Author Response

Thank you for your kind and valuable comments. We have comprehensively revised our manuscript while providing a point-by-point response to the reviewer’s comments. The revised text is highlighted with yellow highlights.

General Comments

This study ‘Rapid extraction of viral nucleic acids using rotating blade lysis and magnetic beads’ seeks to present an alternate method for RNA extraction from viral samples, as a tool for field applications. Although, an alternative approach will be a welcome tool if it is cost effective and rapid, the authors should address these concerns.

Ans) Thank you for your kind comments. Throughout this study, we have focused on creating a cost-effective and rapid method using existing tools that are readily available in a laboratory environment. For example, a blade stirrer can be made of magnetized steel sheet with scissors. Small tubes and rotary stirrers are easily accessible in the laboratory. We have added the corresponding statement on the page 7, line 245.

Method and Results

1) What reagent were the nasal swabs used to test the proposed system stored in?

Ans) The BD Universal Viral Transport Collection kit was used. Specimens were collected using 3 ml VTM, aliquoted into 1.5 ml tubes and stored at -80°C. (on page 3, line 102)

2) A highly pure RNA sample is expected to have a 260/280 ration of 2.0, How do authors explain the seeming lack of purity in the field samples

Ans) In general, the purity of nucleic acid sample can be estimated with the absorbance ration (A260/A280). As reviewer indicated, a highly pure RNA sample used to yield 2.0 for the 260/280 ratio. However, the acceptable ratio of the 260/280 range for pure RNA ranges 1.6 - 2.0.[1] The absorbance ratios (A260/A280) of all samples in this study ranged from 1.6 to 2.0. We've added a reference to this. (on page 6, line 203)

3) How did the ratio (purity) of RNA extracted using the proposed method compare with that of the Kit?

Ans) As shown in Fig. 5(a), we compared the absorbance ratios (A260/A280) of three samples between QIAamp and the proposed method. All samples of the proposed method ranged within 1.6 and 2.0, whereas one sample of the QIAamp kit was 1.4, out of the optimal absorbance ratios (A260/A280).

4) It is not enough for authors to show only an agarose gel image for the field samples, it will be necessary for authors to sequence the amplified product and confirm that it is indeed influenza.

Ans) Thank you for your valuable suggestion for confirmation whether the amplified product was the targeted influenza virus or not. We have described primer sequences for the use of RT-qPCR, which precisely targets the housekeeping gene (M gene, 154 bp) of the influenza A virus, as suggested by WHO.[2] As depicted in Figs.2-5, all the gel analysis results showed that the amplified products yield the length of 154 bp as we designed. Furthermore, throughout the entire present study, the target gene was amplified and confirmed with RT-qPCR, which false amplification rate is less than 0.1%. Thus, we think that there is no need for sequence the amplified products in this study. We’ve added a reference to this. (on page 4, line 136)

5) Do authors expect the proposed method to be applicable to other biological samples such as blood and or arthropods?

Ans) Thanks for the great advice. This method is the principle of breaking down small molecules such as viruses in liquids by physical lysing methods. It would be possible to extend the current method to other types of samples as long as the existing proteins in the sample do not prevent the nucleic acids from adsorbing to magnetic beads. As a follow-up experiment, With the present method, we are examining the possibility to extract exosomal miRNA from plasma without exosome isolation step. Thus, we have added the corresponding statement on page 7, line 259.

Reference :

  1. Yu, Y. J., Majumdar, A. P., Nechvatal, J. M., Ram, J. L., Basson, M. D., Heilbrun, L. K., & Kato, I. (2008). Exfoliated cells in stool: A source for Reverse Transcription-PCR–based analysis of biomarkers of gastrointestinal cancer. Cancer Epidemiology Biomarkers & Prevention, 17(2), 455-458.
  2. World Health Organization. (2017). WHO information for the molecular detection of influenza viruses. Updated July2017.

Reviewer 2 Report

In the manuscript by Bae et al., the authors reported a rapid viral RNA extraction method using physical lysis with rotating blade and isolated with magnetic beads. The viral membrane was instantly lysed using a high-speed rotating blade, released RNA was immediately absorbed with magnetic silica surface beads. The authors combined the two steps into one step in a single tube. The authors first determined the optimum rotational speed of blade to 4000 rpm and then found 1 min rotation time is enough for lyse. After that, the authors compared the two-step protocol and the single step protocol (include magnetic beads in the rotating blade system) and found single step protocol minimized the loss of released RNA. Finally, the authors applied the single step protocol to three clinical samples and found their single step protocol got comparable performance to the commercial QIAamp Viral RNA Mini kit.

Overall, it is an interesting study and presented a simple and fast viral RNA extract method. However, I have a few questions.

1, What’s the size of RNA tested in the studies? Is the method reported applied to all size of RNAs? Or is there any size limitation? As the blade may cut longer RNA more efficiently.

2, Did the authors test this method with DNA extraction as the title has the words “Rapid extraction of Viral Nucleic Acids” instead of “RNA”?

3, Page 3,  Line 93, please check if the text font is consistent or not.

Author Response

Thank you for your kind and valuable comments. We have comprehensively revised our manuscript while providing a point-by-point response to the reviewer’s comments. The revised text is highlighted with yellow highlights.

1) What’s the size of RNA tested in the studies? Is the method reported applied to all size of RNAs? Or is there any size limitation? As the blade may cut longer RNA more efficiently.

Ans) Thank you for your valuable question. In fact, we have considered the potential risk of RNA cutting due to the rotational blade.

First, the total RNA length of the influenza virus A we targeted was 13.5 kb. Second, we have not examined the size limitation whether the present method is applicable to all size of viral RNA. At least, we confirmed that short RNA such as miRNA in exosomes was efficiently extracted from plasma with a single step. Third, we have confirmed in a separate study (not reported) that a rotating blade without magnetic beads can efficiently cut DNA and RNA. However, it is worth of noting that the rotational blade was aimed for lysing viral membrane rather than nucleic acids.

2) Did the authors test this method with DNA extraction as the title has the words “Rapid extraction of Viral Nucleic Acids” instead of “RNA”?

Ans) Thank you for your sharp point. As you know, most of viruses are RNA virus type but some of them are DNA virus such as herpes, smallpox, hepatitis B, adenoviruses, and warts. Thus, we want to extend our study for DNA virus with the proposed method. Also, we are in the middle of conducting experiments of DNA and miRNA extraction from exosomes, which are known as a treasure chest of nucleic acids in plasma. We have added the corresponding statement in the text and references. (on page 7, line 259)

3) Page 3,  Line 93, please check if the text font is consistent or not.

Ans) Thanks for pointing out. We have confirmed.